# Wavelet Feature Maps Compression for Image-to-Image CNNs

**Shahaf E. Finder** *, **Yair Zohav** *, **Maor Ashkenazi** *, **Eran Treister**
The Department of Computer Science, Ben-Gurion University
[finders,maorash]@post.bgu.ac.il   erant@cs.bgu.ac.il

## Abstract

Convolutional Neural Networks (CNNs) are known for requiring extensive computational resources, and quantization is among the best and most common methods for compressing them. While aggressive quantization (i.e., less than 4-bits) performs well for classification, it may cause severe performance degradation in image-to-image tasks such as semantic segmentation and depth estimation. In this paper, we propose Wavelet Compressed Convolution (WCC)—a novel approach for high-resolution activation maps compression integrated with point-wise convolutions, which are the main computational cost of modern architectures. To this end, we use an efficient and hardware-friendly Haar-wavelet transform, known for its effectiveness in image compression, and define the convolution on the compressed activation map. We experiment with various tasks that benefit from high-resolution input. By combining WCC with light quantization, we achieve compression rates equivalent to 1-4bit activation quantization with relatively small and much more graceful degradation in performance. Our code is available at https://github.com/BGUCompSci/WaveletCompressedConvolution.

## 1 Introduction

Over the past years, Convolutional Neural Networks (CNNs) have brought significant improvement in processing images, video, and audio [35, 34]. However, CNNs require significant computational and memory costs, which makes the usage of CNNs difficult in applications where computing power is limited, *e.g.*, on edge devices. To address this limitation, several approaches have been proposed to reduce the computational costs of neural networks. Among the popular ones are weight pruning [25, 23], architecture search [27, 33], and quantization [36, 4]. In principle, all these approaches can be applied simultaneously on top of each other to reduce the computational costs of CNNs.

Specifically, the quantization approach relieves the computational cost of CNNs by quantizing their weights and activation (feature) maps using low numerical precision so that they can be stored and applied as fixed point integers [29, 4]. In particular, it is common to apply aggressive quantization (less than 4-bit precision) to compress the activation maps [18]. However, it is known that compressing natural images using uniform quantization is sub-optimal. Indeed, applying aggressive quantization in certain CNNs can lead to significant degradation in the network's performance. The impact is especially evident for image-to-image tasks such as semantic segmentation [55] and depth prediction [22], where each pixel has to be assigned a value. *E.g.*, in a recent work that targets quantized U-Nets [55], activations bit rates are kept relatively high ($\sim 8$ bits) while the weight bit rates are lower (down to 2 bits). Beyond that, we note that the majority of the quantization works are applied and tested on image classification [18, 37, and references therein] and rarely on other tasks.

---

*Contributed equally.

36th Conference on Neural Information Processing Systems (NeurIPS 2022).

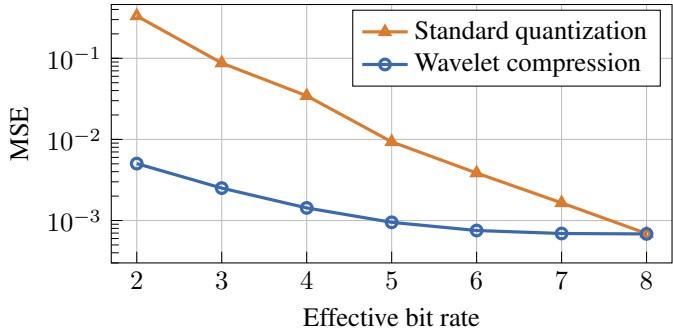

Figure 1: Comparison of standard quantization and our proposed wavelet compression. The ordinate represents MSE between the quantized and original activation maps based on $10^3$ activation maps from the second hidden layer of MobileNetV3 (small) using the ImageNet data set.

This work aims to improve the compression of the activation maps by introducing Wavelet Compressed Convolution (WCC) layers, which use wavelet transforms to compress activation maps before applying convolutions. To this extent, we utilize Haar-wavelet transform [13] due to our ability to apply it (and its inverse) efficiently in linear complexity for each channel, using additions and subtractions only, thanks to the simplicity of the Haar basis. The core idea of our approach is to keep the same top $k$ entries in magnitude of the transformed activation maps with respect to *all channels* (dubbed as joint shrinkage) and perform the convolution in the wavelet domain on the *compressed* signals, saving significant computations. We show that the transform and shrinkage operations commute with the $1 \times 1$ (point-wise) convolution, the heart of modern CNNs. This procedure is applied along with modest quantization to reduce computational costs further.

To summarize, our key contributions are: 1) We propose Wavelet Compressed Convolution (WCC), a novel approach to compress $1 \times 1$ convolutions using a modified wavelet compression technique. 2) We demonstrate that applying low-bit quantization on popular image-to-image CNNs may yield degradation in performance. Specifically, we show that for object detection, semantic segmentation, depth prediction, and super-resolution. 3) We show that using WCC dramatically improves the results for the same compression rates, using it as a drop-in replacement for the $1 \times 1$ convolutions in the baseline network architectures.

## 2 Related Work

**Quantized neural networks.** Quantized neural networks have been quite popular recently and are exhibiting impressive progress in the goal of true network compression and efficient CNN deployment. Quantization methods include [70, 69, 5], and in particular [37, 18, 11], which show that the clipping parameters—an essential parameter in quantization schemes—can be learned through optimization. Beyond that, there are more sophisticated methods to improve the mentioned schemes. For example, dynamic quantization schemes utilize different bit allocations at every layer [14, 9, 57]. Non-uniform methods can improve the quantization accuracy [66] but require a look-up table, which reduces hardware efficiency. Quantization methods can also be enhanced by combination with pruning [56] and knowledge distillation for better training [31].

The works above focus on image classification. When considering image-to-image tasks (*e.g.* semantic segmentation) networks tend to be more sensitive to quantization of the activations. In the work of [2], targeting segmentation of medical images, the lowest bit rate for the activations is 4 bits, and significant degradation in the performance is evident compared to 6 bits. These results are consistent with the work of [55] that was mentioned earlier, which uses a higher bit rate for the activations than for the weights. [65] use weight (only) quantization for medical image segmentation as an attempt to remove noise and not for computational efficiency. The recent work of [41] shows both a sophisticated post-training quantization scheme and includes fine-tuned semantic segmentation results. Again a significant degradation is observed when going from 6 to 4 bits. One exception is the work of [26] that uses ternary networks (equivalent to 2 bits here) and segments one medical data set relatively well compared to its full-precision baseline.

In this work, we focus on the simplest possible quantization scheme: uniform (across all weights and activations), quantization-aware training, and per-layer clipping parameters. That is to ensure hardware compatibility and efficient use of available training data. In principle, one can run our platform regardless of the quantization type and scenario (*e.g.*, non-uniform/mixed precision quantization). Also, since we target activations' compression, any method focused on the weights (*e.g.*, pruning [56]) can also be combined with our proposed method.

**Wavelet transforms in neural networks.** Wavelet transforms are widely used in image processing [48]. For example, the JPEG2000 format [49] uses the wavelet domain to represent images as highly sparse feature maps. Recently, wavelet transforms have been used to define convolutions and architectures in CNNs for various imaging tasks: [20] incorporate wavelets in Generative Adversarial Networks to enhance the visual quality of generated images as well as improve computational performance; [28] present a network architecture for super-resolution, where the wavelet coefficients are predicted and used to reconstruct the high-resolution image; [15] and [62] use wavelet transforms in place of pooling operators to improve CNNs performance. The former uses a dual-tree complex wavelet transform [32], and the latter learns the wavelet basis as part of the network optimization. In all of these cases, the wavelet transform is not used for compression but rather to preserve information with its low-pass filters. [40] suggest using a modified U-Net architecture for image-to-image translation. There, wavelet transforms are used for down-sampling, and the inverse is used for up-sampling. This work is architecture-specific, and the method can not be easily integrated into other architectures. In contrast, our proposed WCC layer can easily replace $1 \times 1$ convolutions regardless of the CNN architecture. Hence our framework is, in principle, also suitable for post-training quantization [6], where the data is unavailable, and the original network is not retrained.

In the context of compression, [63] propose a wavelet-based approach to learn basis functions for the wavelet transform to compress the weights of linear layers, as opposed to compression of the activation as we apply here. We use the Haar transform (as opposed to a learned one) for its hardware efficiency. Using a different transform (known or learned) is also possible at the corresponding computational cost. [51] introduce quantization in the wavelet domain, as we do in this work. However, the authors suggest improving the quantization scheme by learning a different clipping parameter per wavelet component, but without the feature shrinkage stage, which is the heart of our approach (we use the same clipping parameter for the whole layer using 8 bits for hardware efficiency). As mentioned before, one can use our WCC together with different types of quantization schemes, including the one proposed by [51], taking into account the additional hardware complexity of using different clipping parameters per component.

## 3    Background

**Quantization-aware training.** Quantization schemes can be divided into post- and quantization-aware training schemes. Post-training schemes perform model training and model quantization separately, which is most suitable when the training data is unavailable during the quantization phase [6, 45, 8]. On the other hand, quantization-aware training schemes are used to adapt the model's weights as an additional training phase. Such schemes do require training data but generally provide better performance. Quantization schemes can also be divided into uniform vs. non-uniform methods, where the latter is more accurate, but the former is more hardware-friendly [37, 30]. Lastly, quantization schemes can utilize quantization parameters per-channel within each layer or utilize these parameters only per layer (where all the channels share the same parameters). Similar to before, per-channel methods are difficult to exploit in hardware, while per-layer methods are less accurate but more feasible for deployment on edge devices. This paper focuses on per-layer and uniform quantization-aware training for both weights and activations and aims to improve on top of it. Other quantization schemes can be applied within our wavelet compression framework as well.

In quantization-aware training, we set the values of the weights to belong to a small set so that after training, the application of the network can be carried out in integer arithmetic, *i.e.*, activation maps are quantized as well. Even though we use discontinuous rounding functions throughout the network, quantization-aware training schemes utilize gradient-based methods to optimize the network's weights [25, 68]. In a nutshell, when training, we iterate on the floating-point values of the weights. During the forward pass, both the weights and activation maps are quantized, while during the backward pass, the Straight Through Estimator (STE) approach is used [7], where we ignore the rounding function, whose exact derivative is zero.

The specific quantization scheme that we use is based on the work of [37]. First, the pointwise quantization operator is defined by:

$$q_b(t) = \frac{\text{round}((2^b-1)\cdot t)}{2^b-1},$$

(1)

where $t \in [-1, 1]$ or $t \in [0, 1]$ for signed or unsigned quantization, respectively[2]. The parameter $b$ is the number of bits that are used to represent $t$. Each entry undergoes three steps during the forward pass: scale, clip, and round. That is, to get the quantized version of a number, we apply:

$$x_b = \begin{cases} \alpha q_{b-1}(\text{clip}(\frac{x}{\alpha}, -1, 1)) & \text{if signed} \\ \alpha q_b(\text{clip}(\frac{x}{\alpha}, 0, 1)) & \text{if unsigned} \end{cases}.$$

(2)

Here, $x, x_b$ are the real-valued and quantized tensors, and $\alpha$ is the clipping parameter. The parameters $\alpha$ in (2) play an important role in the error generated at each quantization and should be chosen carefully. Recently, [37, 18] introduced an effective gradient-based optimization to find the clipping values $\alpha$ for each layer, again using the STE approximation. This enables the quantized network to be trained in an end-to-end manner with backpropagation. To further improve the optimization, weight normalization is also used before each quantization.

**Haar wavelet transform and compression.** In this section, we describe the Haar-wavelet transform in deep learning language and its usage for compression. See [59] for more details on the use of wavelets for image compression. Given an image channel $\mathbf{x}$, the one-level Haar transform can be achieved by a separable 2D convolution with a stride of 2 using the following weight tensor:

$$\mathbf{W} = \frac{1}{2} \left[ \begin{bmatrix} 1 & 1 \\ 1 & 1 \end{bmatrix}, \begin{bmatrix} 1 & -1 \\ 1 & -1 \end{bmatrix}, \begin{bmatrix} 1 & 1 \\ -1 & -1 \end{bmatrix}, \begin{bmatrix} 1 & -1 \\ -1 & 1 \end{bmatrix} \right].$$

(3)

These kernels can also be expressed as a composition of separable 1D kernels $[1, 1]/\sqrt{2}$ and $[1, -1]/\sqrt{2}$. The result of the convolution $[\mathbf{y}_1, \mathbf{y}_2, \mathbf{y}_3, \mathbf{y}_4] = \text{Conv}(\mathbf{W}, \mathbf{x})$ has four channels, each of which has half the resolution of $\mathbf{x}$. The leftmost kernel in (3) is an averaging kernel, and the three right kernels are edge-detectors. This, together with the fact that images are piece-wise smooth, leads to relatively sparse images $\mathbf{y}_2, \mathbf{y}_3, \mathbf{y}_4$. Hence, if we retain only the few top-magnitude entries in these vectors, we keep most of the information, as most of the entries we drop are zeros. That is the main idea of wavelet compression. We denote the Haar-wavelet transform by $\mathbf{y} = \text{HWT}(\mathbf{x})$, where $\mathbf{y}$ is defined as the concatenation of the vectors $\mathbf{y}_1, ..., \mathbf{y}_4$ into one. Since the kernels in (3) form an orthonormal basis, applying the inverse transform is obtained by the transposed convolution of (3):

$$\mathbf{x} = \text{iHWT}(\mathbf{y}) = \text{Conv-transposed}(\mathbf{W}, [\mathbf{y}_1, \mathbf{y}_2, \mathbf{y}_3, \mathbf{y}_4]).$$

(4)

However, unlike $\mathbf{y}_2, \mathbf{y}_3, \mathbf{y}_4$, the image $\mathbf{y}_1$ is not sparse, and is just the down-sampled $\mathbf{x}$. Hence, in the multilevel wavelet transform, we recursively apply the convolution with $\mathbf{W}$ on low-pass filtered sub-bands $\mathbf{y}_1$ to generate further down-sampled sparse channels. For example, a 2-level Haar transform can be summarized as

$$\text{Conv}(\mathbf{W}, \mathbf{x}) = \left[\mathbf{y}_1^1, \mathbf{y}_2^1, \mathbf{y}_3^1, \mathbf{y}_4^1\right]; \quad \text{Conv}(\mathbf{W}, \mathbf{y}_1^1) = \left[\mathbf{y}_1^2, \mathbf{y}_2^2, \mathbf{y}_3^2, \mathbf{y}_4^2\right],$$

(5)

and the resulting transformed image with 2-levels can be written as the concatenated vector

$$\text{HWT}(\mathbf{x}) = \mathbf{y} = \left[\mathbf{y}_1^2, \mathbf{y}_2^2, \mathbf{y}_3^2, \mathbf{y}_4^2, \mathbf{y}_2^1, \mathbf{y}_3^1, \mathbf{y}_4^1\right].$$

(6)

In this work we use 3 levels in all the experiments. To apply compression we define the operator $\mathbf{T}$ that extracts the top $k$ entries in magnitude from the vector $\mathbf{y}$:

$$\mathbf{y}^{compressed} = \mathbf{T} \cdot \text{HWT}(\mathbf{x}).$$

(7)

To de-compress this vector we first zero-fill $\mathbf{y}^{compressed}$ (*i.e.*, multiply with $\mathbf{T}^\top$) and apply the inverse Haar transform, which involves with the transposed operations in the opposite order.

## 4 Wavelet Compressed Convolution

In this work, we aim to reduce the memory bandwidth and computational cost associated with convolutions performed on intermediate activation maps. To this end, we apply the Haar-wavelet

---

[2] In most standard CNNs, the ReLU activation is used; hence the activation feature maps are non-negative and can be quantized using an unsigned scheme. If a different activation function that is not non-negative is used, or, as in our case, the wavelet coefficients are quantized, signed quantization should be used instead.

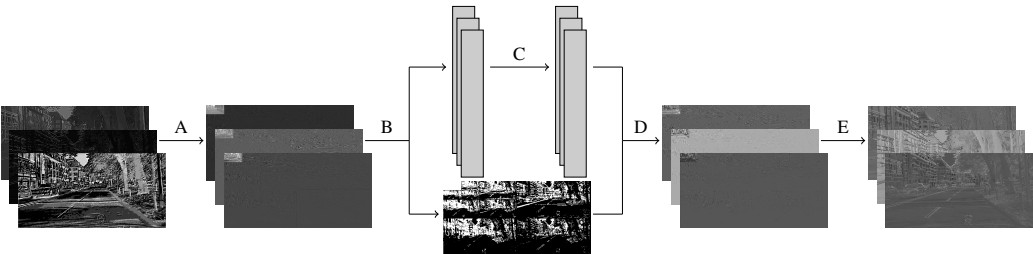

Figure 2: The workflow of WCC on a 3-channel input from the Cityscapes dataset. **A**: Haar-wavelet transform; **B**: Joint shrinkage, transforming the input into equally sized 1D vectors and a single bit-map to represent the spatial location of the top $k$ entries; **C**: $1 \times 1$ convolution over the 1D vectors; **D**: Inverse shrinkage (zero filling); **E**: Inverse Haar transform.

transform to compress the activation maps, in addition to light quantization of 8 bits. Our method is most efficient for scenarios with high-resolution feature maps (*i.e.*, large images, whether in 2D or 3D), where the wavelet compression is most effective. Such cases are mostly encountered in image-to-image tasks like semantic segmentation, depth prediction, image denoising, in-painting, super-resolution, and more. Typically, in such scenarios, the size and memory bandwidth used for the weights are relatively small compared to those used for the features.

**Convolution in the wavelet domain.** We focus on the compression of fully-coupled $1 \times 1$ convolutions, as these are the workhorse of lightweight and efficient architectures like MobileNet [50], ShuffleNet [44], EfficientNet [52], ResNeXt [64], and ConvNext [42]. All these modern architectures rely on $1 \times 1$ point-wise convolutions in addition to grouped or depthwise spatial convolutions (*i.e.*, with $3 \times 3$ or larger kernels), which comprise a small part of the computational effort in the network—the $1 \times 1$ operations dominate the inference cost (see Appendix A and Appendix C). Since we focus on computational efficiency, we use the Haar transform, as it is the simplest and most computationally efficient wavelet variant. Indeed, the Haar transform involves binary pooling-like operations, which include only additions, subtractions, and a bit-wise shift. Other types of wavelets are also suitable in our framework while considering the corresponding computational costs—we demonstrate this point in Appendix F.

The main idea of our method is to transform and compress the input using the wavelet transform prior to the $1 \times 1$ convolution, then apply it in the wavelet domain on a fraction of the input size. Since the wavelet compression is applied separately on each channel, it commutes with the $1 \times 1$ convolution. Hence, applying the convolution in the wavelet domain is equivalent to applying it in the spatial domain.

More precisely, the advantage of the wavelet transforms is their ability to compress images. Denote the Haar transform operator as $\mathbf{H}$, *i.e.*, $\mathbf{Hx} = \text{HWT}(\mathbf{x})$. Then, for most natural images $\mathbf{x}$ we have

$$\mathbf{x} \approx \mathbf{H}^\top \mathbf{T}^\top \mathbf{THx}, \tag{8}$$

where $\mathbf{T}$ is the shrinkage operator in Eq. (7) (top $k$ extractor). Because of its orthogonality, $\mathbf{H}^\top$ is the inverse transform iHWT. Our WCC layer is defined by:

$$\text{WCC}(\mathbf{K}_{1\times1}, \mathbf{x}) = \mathbf{H}^\top \mathbf{T}^\top \mathbf{K}_{1\times1} \mathbf{THx}, \tag{9}$$

where here $\mathbf{x}$ is a feature tensor and $\mathbf{K}_{1\times1}$ is a learned $1 \times 1$ convolution operator. The workflow is illustrated in Figure 2, and an explicit algorithm appears in Appendix B. The convolution operates on the compressed domain, hence, if $\mathbf{T}$ can significantly reduce the dimensions of the channels without loosing significant information, this leads to major savings in computations. Note that $\mathbf{H}$ and $\mathbf{T}$ operate on all channels in the same way, *i.e.*, $\mathbf{T}$ extracts the same entries from all channels. We motivate on this next.

**Joint hard shrinkage.** Prior to the $1 \times 1$ convolution, the spatial wavelet transform is applied, and we get sparse feature maps. Since the different channels result from the same input image propagated through the network, they typically include patterns and edges at similar locations. Hence the sparsity pattern of their wavelet-domain representation is relatively similar. This idea of redundancy in the channel space is exploited in the works of [24, 16, 3], where parts of the channels are used to represent the others. Therefore, in $\mathbf{T}$, we perform a joint shrinkage operation between all the channels, in

which we zero and remove the entries with the smallest feature norms across channels, resulting in a compressed representation of the activation maps[3]. The locations of the non-zeros in the original image are kept in a *single* index list or a bit-map for all the channels in the layer, as they are needed for the inverse transform back to the spatial domain. We also apply light 8-bit quantization to the transformed images to improve the compression rate further. The weights and wavelet-domain activations are quantized using the symmetric scheme described in section 3.

**Equivalence of the convolutions.** Our method aims at compression only. Hence, we show that a $1 \times 1$ convolution kernel can be applied both in the spatial and in the compressed wavelet domain. By its definition, we can write a $1 \times 1$ convolution as a summation over channels. That is:

$$\mathbf{y} = \mathbf{K}_{1\times1}\mathbf{x} \Rightarrow \mathbf{y}_i = \sum_j k_{ij}\mathbf{x}_j, \tag{10}$$

where $k_{ij} \in \mathbb{R}$ are the weights of the convolution tensor. Now, suppose we wish to compress the result $\mathbf{y}$, through (8). Because $\mathbf{T}$ and $\mathbf{H}$ are separable and spatial, we get:

$$\mathbf{y}_i \approx \mathbf{H}^\top\mathbf{T}^\top\mathbf{T}\mathbf{H}(\mathbf{K}_{1\times1}\mathbf{x})_i = \mathbf{H}^\top\mathbf{T}^\top\sum_j k_{ij}\mathbf{T}\mathbf{H}\mathbf{x}_j, \tag{11}$$

where the exact equality holds assuming that $\mathbf{T}$ extracts the same entries in both cases. Hence,

$$\mathbf{y} \approx \mathbf{H}^\top\mathbf{T}^\top\mathbf{K}_{1\times1}\mathbf{T}\mathbf{H}\mathbf{x}. \tag{12}$$

That is, the wavelet compression operator, which is known to be highly efficient for natural images, commutes with the $1 \times 1$ convolution, so the latter can be applied on the compressed signal and get the same result as compressing the result directly. This introduces an opportunity to save computations on the one hand and use more accurate compression on the other.

The description above is suited for $1 \times 1$ convolutions only, while many CNN architectures involve spatial convolutions with larger kernels, strides, etc. The main concept is that fully coupled convolutions that mix all channels are expensive, inefficient, and redundant when used with large spatial kernels [17]. The spatial mixing can be obtained using separable convolutions at less cost without losing efficiency (see Appendix A). Since we aim at saving computations, separating the kernels in the architecture is recommended even before discussing any form of lossy compression [10, 50]. Furthermore, separable convolutions (and the Haar transform) can be applied separately in chunks of channels, and the memory bandwidth can be reduced. The $1 \times 1$ convolution, on the other hand, involves all channels at once, and it is harder to reduce the bandwidth in this case. With our approach, the input for the convolution is compressed, and the bandwidth is reduced (see Appendix H).

**Limitations.** Our approach is best suited for applications where feature maps are of high resolution. This includes various computer vision tasks, as we demonstrate in section 5. The majority of other works show only classification results, which is a less sensitive task than others for the input's resolution. For example, for ImageNet, images are typically resized to $224 \times 224$ and down-sampled aggressively to $7 \times 7$ using a few layers. For such a low resolution, the wavelet shrinkage is not as effective as it is for high-resolution feature maps. On the other hand, quantization by itself is already very effective for classification, hence tackling other tasks in this context may be highly beneficial.

## 5   Results

This section evaluates our proposed WCC layer's performance on four tasks: object detection, semantic segmentation, monocular depth estimation, and super-resolution. The majority of the feature maps in these tasks are relatively large and very well-suited for wavelet compression. We use quantization-aware training based on the work of [37] for the baseline with and without WCC.

As mentioned in section 2, most quantization works evaluate their performance on image classification. An exception is the work of [46], which shows results for object detection and semantic segmentation, although we were unable to reproduce their results. Therefore, when comparing with their work, we measure the effectiveness of WCC in the relative degradation of the score when compressing the activation further than 8-bit; this is due to the absolute accuracy being strongly dependent on the baseline quantization mechanism and training process.

---

[3]Regardless of the joint sparsity of the channels, some approaches suggest taking the left-upmost part of the wavelet transform for any image, so in principle, the joint sparsity may suffice for a general set of images as well.

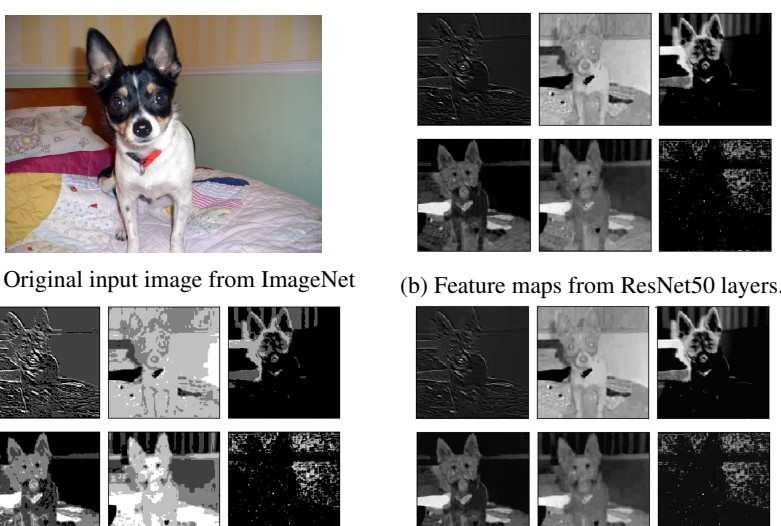

(a) Original input image from ImageNet

(b) Feature maps from ResNet50 layers.

(c) 2-bit quantized layers (unsigned).

(d) 2-bit equivalent wavelet compressed.

Figure 3: Feature maps from layers 2 and 3 (top and bottom triplets, respectively) of a pre-trained ResNet50 on ImageNet. The maps are compressed with uniform quantization (2-bit) and wavelet compression (25% shrinkage + 8-bit quantization, equivalent to 2-bit quantization). Clearly, the wavelet compression loses much less information than aggressive quantization.

The general training scheme used in the subsequent experiments is similar to [37]. We either start with full-precision pre-trained network weights or train a full-precision network to convergence. Afterward, we gradually reduced the bit rates and the WCC compression factor. We detail the rest of our experimental setup in each section separately. In each network presented in the results, the $1 \times 1$ convolution layers comprise 85% to 90% of the total operations (*e.g.*, see Appendix C). We implemented our code using PyTorch [47], based on Torchvision and public implementations of the chosen networks. We ran our experiments on NVIDIA 24GB RTX 3090 GPU. The computational cost of each model is measured in Bit-Operations (BOPs, see Appendix D).

## 5.1 Qualitative Compression Assessment

First, we qualitatively demonstrate the advantage of our approach. We compare standard quantization's mean square error (MSE) to our proposed method on a feature map from a pre-trained MobileNetV3 applied on a batch of $10^3$ images from the ImageNet dataset [34]. We consider an effective bit rate range of $[2, 8]$. The bit rate of the shrunk wavelet coefficients is 8; multiplying it with the compression ratio yields the effective bit rate (*e.g.*, 8 bit and 25% shrinkage is equivalent to 2 bit). Figure 1 shows the MSE comparison per effective bit rate, and Figure 3 presents a visualization comparison of a typical activation map compressed by a standard quantization and our method. Both show that the information loss is more significant in standard quantization.

## 5.2 Object Detection

We apply our method to EfficientDet [54] using the EfficientDet-D1 variant. We train and evaluate the networks on the MS COCO 2017 [39] object detection dataset. The dataset contains 118K training images and 5K validation images of complex everyday scenes, including detection annotations for 80 classes of common objects in their natural context. Images are of size $\sim 640 \times 480$ pixels, and as defined for the D1 variant, are resized to $640 \times 640$ pixels. We use a popular publicly available PyTorch EfficientDet implementation[4] and adopt the training scheme suggested by its authors. We use the AdamW optimizer, with a learning rate of $10^{-3}$ when initially applying WCC layers and $10^{-4}$ for finetuning. In addition, we apply a learning rate warm-up in the first epoch of training, followed by a cosine learning rate decay. Each compression step is finetuned for 20 to 40 epochs. The results

---

[4]`https://github.com/rwightman/efficientdet-pytorch`

Table 1: Validation results on MS COCO using EfficientDet-D1. Degradation from the baseline is in parentheses.

| Precision (W/A) | Wavelet shrinkage | BOPs (B) | ↑ mAP |
|---|---|---|---|
| FP32 | None | 6,144 | 40.08 |
| [46] | | | |
| 4bit / 8bit | None | 280.4 | 35.34 |
| 4bit / 4bit | None | 185.8 | 24.70 (-10.64) |
| Our baseline + WCC | | | |
| 4bit / 8bit | None | 280.4 | 31.44 |
| 4bit / 8bit | 50% | 198.5 | 31.15 (-0.29) |
| 4bit / 8bit | 25% | 155.4 | 27.49 (-3.95) |

Table 2: Validation results for Pascal VOC using DeepLabV3plus(MobileNetV2). Degradation from the baseline is in parentheses.

| Precision (W/A) | Wavelet shrinkage | BOPs (B) | ↑ mIoU |
|---|---|---|---|
| [46] | | | |
| FP32 | None | 4,534 | 0.729 |
| 4bit / 8bit | None | 141 | 0.709 |
| 4bit / 4bit | None | 70 | 0.668 (-0.041) |
| Our baseline + WCC | | | |
| FP32 | None | 4,534 | 0.715 |
| 4bit / 8bit | None | 141 | 0.675 |
| 4bit / 8bit | 50% | 76 | 0.661 (-0.014) |
| 4bit / 8bit | 25% | 42 | 0.583 (-0.092) |
| 4bit / 8bit | 12.5% | 24 | 0.515 (-0.160) |

are presented in Table 1. We train a quantized network using 4-bit weights and 8-bit activations, used as a baseline, and then further compress the activations using WCC with a 50% compression factor, which results in comparable BOPs of a 4bit/4bit quantized network. The result significantly surpasses the accuracy achieved by [46], both in the absolute score and in relative degradation from the baseline. We continue to show that even when applying a 25% compression factor, our method outperforms the 4bit/4bit network of [46] while reducing the BOPs significantly.

### 5.3 Semantic Segmentation

For this experiment, we use the popular DeeplabV3plus [10] with a MobileNetV2 backbone [50]. We evaluated our proposed method on the Cityscapes and Pascal VOC datasets. The Cityscapes dataset [12] contains images of urban street scenes; the images are of size $1024 \times 2048$ with pixel-level annotation of 19 classes. During training, we used a random crop of size $768 \times 768$ and no crop for the validation set. The Pascal VOC [19] dataset contains images of size $513 \times 513$ with pixel-level annotation of 20 object classes and a background class. We augmented the dataset similarly to [10]. We used two configurations for the weights—4 and 8 bits—and for each, we used various compression rates for the activations, both for the standard quantization and WCC.

For optimization, we used SGD with momentum $0.9$, weight decay $10^{-4}$, learning rate decay $0.9$, and batch size $16$. For Cityscapes, we trained the full-precision model for 160 epochs with a base learning rate of $10^{-1}$. Each compression step was finetuned for 50 epochs with a learning rate of $10^{-2}$. For Pascal VOC, we trained the full-precision model for 50 epochs with a learning rate of $10^{-2}$. Each compression step was finetuned for 25 epochs with a base learning rate of $2 \cdot 10^{-3}$.

Table 2 presents the results for Pascal VOC. In relative degradation, [46] experienced a drop of $\sim$0.041 between 4bit/8bit and 4bit/4bit. In contrast, our method experiences a $\sim$0.014 drop when applying 50% compression, achieving an equivalent score while using a worse baseline. Figure 4 and Figure 5 compare the results for Cityscapes with different compression configurations. Note that we show compression rates that are computationally equal to

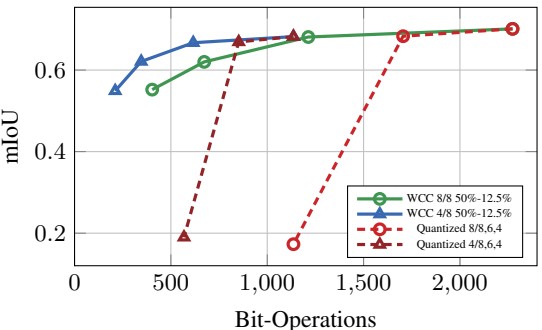

Figure 4: Validation results for Cityscapes using DeepLabV3plus(MobileNetV2) compared to BOPs for different bit-rate quantizations and WCC configurations.

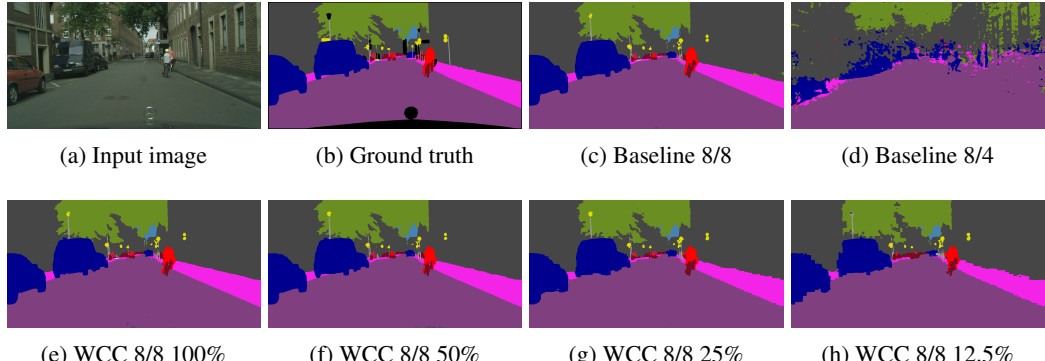



(a) Input image     (b) Ground truth     (c) Baseline 8/8     (d) Baseline 8/4

(e) WCC 8/8 100%     (f) WCC 8/8 50%     (g) WCC 8/8 25%     (h) WCC 8/8 12.5%



Figure 5: Cityscapes segmentation results. All networks use weight quantization of 8-bits. (a) input image. (b) ground truth. (c), (d) normal quantization with 8- and 4-bits activations respectively. (e)-(h) WCC with 8-bits activations and shrinkage rate of 100%, 50%, 25% and 12.5% respectively.

2- and 1-bit, which degrade gracefully. See Appendix E for the detailed results, and Appendix F for additional results with different types of wavelets.

## 5.4 Monocular Depth Estimation

We apply our method to Monodepth2 [22], a network architecture for monocular depth estimation—a task of estimating scene depth using a single image. We evaluated the results on the KITTI dataset [21], containing images of autonomous driving scenarios, each of size $\sim$1241$\times$376 pixels. The train/validation split is the default selected by Monodepth2 (based on [71]), and we evaluate it on the ground truths provided by the KITTI depth benchmark.

We extended the code base to support Mo-bileNetV2 as a backbone, and adapted the depth decoder to use depthwise-separable convolutions for an extra compression factor, without harming the accuracy. All the layers, except the first and final convolutions are compressed. We use the AdamW optimizer, a learning rate of $1.4 \cdot 10^{-4}$, a weight decay of $10^{-3}$ and run each experiment for 20 epochs.

Table 3 presents a comparison between standard quantization and WCC for Monodepth2. Using WCC, we compress the network to very low bit-rates while having a relatively minimal and more graceful performance degradation. Using WCC layers with 50% compression resulted in comparable scores to the

Table 3: Monodepth2(MobileNetV2) results measured by AbsRel and RMSE.

| Precision (W/A) | Wavelet shrinkage | BOPs (B) | ↓ AbsRel | ↓ RMSE |
|---|---|---|---|---|
| FP32 | None | 1,163.6 | 0.093 | 4.022 |
| 8bit / 8bit | None | 133.6 | 0.092 | 4.018 |
| 8bit / 4bit | None | 99.26 | 0.097 | 4.166 |
| 8bit / 2bit | None | 82.1 | 0.268 | 8.223 |
| 8bit / 8bit | 50% | 103.9 | 0.098 | 4.217 |
| 8bit / 8bit | 25% | 88.5 | 0.112 | 4.663 |
| 8bit / 8bit | 12.5% | 80.8 | 0.131 | 5.046 |

alternative 8bit/4bit quantization (with similar BOPs). When applying a compression factor of 25%, and even 12.5%, we achieve superior results to the quantized alternative. A qualitative comparison is presented in Appendix G.

## 5.5 Super-resolution

Another experiment we perform is for the task of super-resolution, which aims to reconstruct a high-resolution image from a single low-resolution image. For this task, we chose the popular EDSR network [38], trained on the DIV2K dataset [1], and used its basic configuration for 2x super-resolution. DIV2K contains images in 2K resolution, and the model is trained on $48 \times 48$ random crops of the down-scaled train images as input (which are not down-scaled further throughout the network). We used the official implementation of EDSR and did not change the training process.

The model consists of three parts: head, body, and tail. In our experience, quantizing the head and tail showed a severe degradation with even the lightest quantization. Therefore, for this experiment, we only compress the body. It is important to mention that this method uses a skip-connection between the head and the tail, so it is to be expected that even aggressive quantization can achieve reasonable results, differently than the other models we show in the paper.

In addition, EDSR uses full $3 \times 3$ convolutions. Hence, we use the technique described in Appendix A, replacing each full-convolution with a set of two convolutions, a depthwise-$3 \times 3$ followed by a pointwise-$1 \times 1$. Empirically, we see no change in results in our experiment following this change, and we could still to replicate the results reported by [38].

Table 4 shows the results for two configurations, WCC without quantization and WCC with light quantization (8/16-bit). The BOPs are calculated for the body's convolutions. The first two rows show our reproduction of [38] results using separable convolutions. The next three rows show the results using our WCC with no quantization, for which the drop in accuracy is minimal. Furthermore, in the quantized results, we can see that adding $50\%$ compression to the 8bit/16bit and 8bit/8bit setups did not influence the accuracy while reducing the costs. Besides that, we again see that WCC achieves better accuracy than quantization alone with aggressive compression (e.g., 8bit/4bit yields a PSNR of 34.49 while 8bit/8bit+50% yields 34.53).

Table 4: EDSR compression results for the task of 2x super-resolution. Note that the first row is the reported result from [38], while the second is our reproduction of it using separable convolutions.

| Precision (W/A) | Wavelet shrinkage | BOPs (B) | ↑ PSNR |
|---|---|---|---|
| FP32/FP32 [38] | None | 975,838 | 35.03 |
| FP32/FP32 | None | 123,674 | 35.02 |
| FP32/FP32 | 50% | 70,017 | 34.98 |
| FP32/FP32 | 25% | 42,910 | 34.93 |
| FP32/FP32 | 12.5% | 29,357 | 34.76 |
| 8bit / 16bit | None | 15,459 | 34.55 |
| 8bit / 8bit | None | 7,730 | 34.53 |
| 8bit / 4bit | None | 3,865 | 34.49 |
| 8bit / 16bit | 50% | 8,961 | 34.55 |
| 8bit / 8bit | 50% | 4,480 | 34.53 |
| 8bit / 8bit | 25% | 2,786 | 34.50 |

## 6   Conclusion

In this work, we presented a new approach for feature map compression, aiming to reduce the memory bandwidth and computational cost of CNNs working at a high resolution. Our approach is based on the classical Haar-wavelet image compression, which has been used for years in standard devices and simple hardware, and in real-time. We save computational costs by applying $1 \times 1$ convolutions on the shrunk wavelet domain together with light quantization. We show that this approach surpasses aggressive quantization using equivalent compression rates.

## Acknowledgments and Disclosure of Funding

The research reported in this paper was supported by the Israel Innovation Authority through the Avatar consortium. The authors also thank the Israeli Council for Higher Education (CHE) via the Data Science Research Center and the Lynn and William Frankel Center for Computer Science at BGU. SF is supported by the Kreitman High-tech scholarship.

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
