# A    A Note on $1 \times 1$ point-wise convolutions

In the case when a certain CNN use $3 \times 3$ convolution only, one can split it to two convolution, a depthwise-$3 \times 3$ and a $1 \times 1$ [60, 58]. Assuming no strides, the depthwise conv involves with $C_{in} \cdot 3 \cdot 3 \cdot N_W \cdot N_H$ MAC operations, while the $1 \times 1$ conv includes $C_{in} \cdot C_{out} \cdot N_W \cdot N_H$ MAC operations ($C_{out}/9$ times more expensive than depthwise). Meaning, for a large enough $C_{out}$, the $3 \times 3$ convolution has about 8-9 times more MAC operations than the depthwise-$3 \times 3$ convolution and $1 \times 1$ convolution.

Some models, such as the ones referenced in section 4, are defined based on that concept. For example, MobilenetV2 consists of residual blocks that perform $1 \times 1$, depthwise-$3 \times 3$, and an additional $1 \times 1$, and for an image input of size $1024 \times 2048$ (*e.g.*, cityscapes), the $1 \times 1$-conv has a MAC count of 18,022M, while the $3 \times 3$-conv has a MAC count of 1,056M (see Appendix C).

A recent paper [42], which shows the impact of Resnet50 modifications, explores the idea of separable convolutions in sections 2.3 and 2.4 and demonstrate the effectiveness of it. For another example, in our Monodepth2 experiment (subsection 5.4), converting to separable convolutions resulted in a 70% drop in BOPs, while AbsRel stayed at 0.093 and RMSE went up from 3.97 to 4.02.

In cases where one might not want to use separable convolutions. We note that the $3 \times 3$ convolution is internally implemented as a matrix-matrix multiplication using different shifts of the image. Hence, the wavelet transform can be adapted to transform the shifted images as well, with a specialized implementation. This implementation might hurt the effectiveness of the joint shrinkage, although for high-resolution images we expect it to behave similarly to the separable convolutions, as large smooth areas are consistent between slightly shifted copies of the same image.

# B    Explicit WCC algorithm

In Alg. 1 below we present a pseudo-code for performing the WCC layer. We note that the Haar wavelet transform can be obtained in-place and there is no real need to allocate new memory for the large intermediate feature maps $X_{ll}^0$ and $Y_{ll}^0$ during WCC. Only the $1 \times 1$ conv operation requires a memory allocation, but it is applied on the shrunken vectors. That is another advantage of WCC as standard conv cannot be applied in-place and needs an allocation of both the large feature maps.

---

**Algorithm 1** Wavelet Compressed Convolution

---

    **Input:** feature map $X \in \mathbb{R}^{n_w \times n_h \times C}$ of spatial size $n_w \times n_h$ and $C$ channels, convolution kernel $K_{1 \times 1}$, wavelet-transform level $d$, compression rate $\gamma$
    $X_{ll}^0 = X$
    **for** $i = 1$ **to** $d$ **do**
        $X_{ll}^i, X_{lh}^i, X_{hl}^i, X_{hh}^i = \text{HWT}(X_{ll}^{i-1})$
    **end for**
    Let $X_{wt}$ be a concatenation of $X_{lh}^i, X_{hl}^i, X_{hh}^i$ for $i = 1 \ldots d$ and $X_{ll}^d$ as in (6)
    Calculate vector norm along the channel dimension of $X_{wt}$.
    Define $I$ as the set of indices of the top $\lceil \gamma n_w n_h \rceil$ vectors by norm.
    $Y_{wt} = Conv(K_{1 \times 1}, X[I])$
    Initialize a zeroed $Y_{ll}^0 \in \mathbb{R}^{n_w \times n_h \times C}$, and set $Y_{ll}^0[I] = Y_{wt}$.
    **for** $i = d - 1$ **to** $0$ **do**
        $Y_{ll}^i = \text{iHWT}(Y_{ll}^{i+1}, Y_{lh}^{i+1}, Y_{hl}^{i+1}, Y_{hh}^{i+1})$.
    **end for**
    Return $Y_{ll}^0$

---

# C MobilenetV2 MACs

A full breakdown of MobilenetV2 MAC operations for a single Cityscapes' image input is provided in Table 5.

Table 5: In depth breakdown of MobilenetV2 (as a backbone for deeplabv3+) for a single $1024 \times 2048$ input. $K$ and $S$ refer to the size of the symmetric kernels and strides respectively. The first convolution of the network is omitted, since it is a common practice to avoid quantizing it.

| Module id | $C_{in}$ | $C_{out}$ | $K$ | Groups | $S$ | Dilation | $H$ | $W$ | MAC |
|---|---|---|---|---|---|---|---|---|---|
| InvRes1 conv1 | 32 | 32 | 3 | 32 | 1 | 1 | 513 | 1025 | 150,552,864 |
| InvRes1 conv2 | 32 | 16 | 1 | 1 | 1 | 1 | 511 | 1023 | 267,649,536 |
| InvRes2 conv1 | 16 | 96 | 1 | 1 | 1 | 1 | 513 | 1025 | 807,667,200 |
| InvRes2 conv2 | 96 | 96 | 3 | 96 | 2 | 1 | 513 | 1025 | 112,914,648 |
| InvRes2 conv3 | 96 | 24 | 1 | 1 | 1 | 1 | 256 | 512 | 301,989,888 |
| InvRes3 conv1 | 24 | 144 | 1 | 1 | 1 | 1 | 258 | 514 | 458,307,072 |
| InvRes3 conv2 | 144 | 144 | 3 | 144 | 1 | 1 | 258 | 514 | 169,869,312 |
| InvRes3 conv3 | 144 | 24 | 1 | 1 | 1 | 1 | 256 | 512 | 452,984,832 |
| InvRes4 conv1 | 24 | 144 | 1 | 1 | 1 | 1 | 258 | 514 | 458,307,072 |
| InvRes4 conv2 | 144 | 144 | 3 | 144 | 2 | 1 | 256 | 514 | 42,467,328 |
| InvRes4 conv3 | 144 | 32 | 1 | 1 | 1 | 1 | 128 | 256 | 150,994,944 |
| InvRes5 conv1 | 32 | 192 | 1 | 1 | 1 | 1 | 130 | 258 | 206,069,760 |
| InvRes5 conv2 | 192 | 192 | 3 | 192 | 1 | 1 | 130 | 258 | 56,623,104 |
| InvRes5 conv3 | 192 | 32 | 1 | 1 | 1 | 1 | 128 | 256 | 201,326,592 |
| InvRes6 conv1 | 32 | 192 | 1 | 1 | 1 | 1 | 130 | 258 | 206,069,760 |
| InvRes6 conv2 | 192 | 192 | 3 | 192 | 1 | 1 | 130 | 258 | 56,623,104 |
| InvRes6 conv3 | 192 | 32 | 1 | 1 | 1 | 1 | 128 | 256 | 201,326,592 |
| InvRes7 conv1 | 32 | 192 | 1 | 1 | 1 | 1 | 130 | 258 | 206,069,760 |
| InvRes7 conv2 | 192 | 192 | 3 | 192 | 2 | 1 | 130 | 258 | 14,155,776 |
| InvRes7 conv3 | 192 | 64 | 1 | 1 | 1 | 1 | 64 | 128 | 100,663,296 |
| InvRes8 conv1 | 64 | 384 | 1 | 1 | 1 | 1 | 66 | 130 | 210,862,080 |
| InvRes8 conv2 | 384 | 384 | 3 | 384 | 1 | 1 | 66 | 130 | 28,311,552 |
| InvRes8 conv3 | 384 | 64 | 1 | 1 | 1 | 1 | 64 | 128 | 201,326,592 |
| InvRes9 conv1 | 64 | 384 | 1 | 1 | 1 | 1 | 66 | 130 | 210,862,080 |
| InvRes9 conv2 | 384 | 384 | 3 | 384 | 1 | 1 | 66 | 130 | 28,311,552 |
| InvRes9 conv3 | 384 | 64 | 1 | 1 | 1 | 1 | 64 | 128 | 201,326,592 |
| InvRes10 conv1 | 64 | 384 | 1 | 1 | 1 | 1 | 66 | 130 | 210,862,080 |
| InvRes10 conv2 | 384 | 384 | 3 | 384 | 1 | 1 | 66 | 130 | 28,311,552 |
| InvRes10 conv3 | 384 | 64 | 1 | 1 | 1 | 1 | 64 | 128 | 201,326,592 |
| InvRes11 conv1 | 64 | 384 | 1 | 1 | 1 | 1 | 66 | 130 | 210,862,080 |
| InvRes11 conv2 | 384 | 384 | 3 | 384 | 1 | 1 | 66 | 130 | 28,311,552 |
| InvRes11 conv3 | 384 | 96 | 1 | 1 | 1 | 1 | 64 | 128 | 301,989,888 |
| InvRes12 conv1 | 96 | 576 | 1 | 1 | 1 | 1 | 66 | 130 | 474,439,680 |
| InvRes12 conv2 | 576 | 576 | 3 | 576 | 1 | 1 | 66 | 130 | 42,467,328 |
| InvRes12 conv3 | 576 | 96 | 1 | 1 | 1 | 1 | 64 | 128 | 452,984,832 |
| InvRes13 conv1 | 96 | 576 | 1 | 1 | 1 | 1 | 66 | 130 | 474,439,680 |
| InvRes13 conv2 | 576 | 576 | 3 | 576 | 1 | 1 | 66 | 130 | 42,467,328 |
| InvRes13 conv3 | 576 | 96 | 1 | 1 | 1 | 1 | 64 | 128 | 452,984,832 |
| InvRes14 conv1 | 96 | 576 | 1 | 1 | 1 | 1 | 66 | 130 | 474,439,680 |
| InvRes14 conv2 | 576 | 576 | 3 | 576 | 1 | 1 | 66 | 130 | 42,467,328 |
| InvRes14 conv3 | 576 | 160 | 1 | 1 | 1 | 1 | 64 | 128 | 754,974,720 |
| InvRes15 conv1 | 160 | 960 | 1 | 1 | 1 | 1 | 68 | 132 | 1,378,713,600 |
| InvRes15 conv2 | 960 | 960 | 3 | 960 | 1 | 2 | 68 | 132 | 70,778,880 |
| InvRes15 conv3 | 960 | 160 | 1 | 1 | 1 | 1 | 64 | 128 | 1,258,291,200 |
| InvRes16 conv1 | 160 | 960 | 1 | 1 | 1 | 1 | 68 | 132 | 1,378,713,600 |
| InvRes16 conv2 | 960 | 960 | 3 | 960 | 1 | 2 | 68 | 132 | 70,778,880 |
| InvRes16 conv3 | 960 | 160 | 1 | 1 | 1 | 1 | 64 | 128 | 1,258,291,200 |
| InvRes17 conv1 | 160 | 960 | 1 | 1 | 1 | 1 | 68 | 132 | 1,378,713,600 |
| InvRes17 conv2 | 960 | 960 | 3 | 960 | 1 | 2 | 68 | 132 | 70,778,880 |
| InvRes17 conv3 | 960 | 320 | 1 | 1 | 1 | 1 | 64 | 128 | 2,516,582,400 |
| Total of $1 \times 1$ | | | | | | | | | 18,022,413,312 |
| Total of $3 \times 3$ | | | | | | | | | 1,056,190,968 |

# D  Computational Costs in Bit Operations (BOPs)

To evaluate the computational cost involved in WCC we use the measure of Bit-Operations (BOPs) [61, 43]. First, the number of Multiply-And-Accumulate (MAC) operations in a convolutional layer is given by

$$\text{MAC(conv)} = C_{\text{in}} \cdot C_{\text{out}} \cdot N_W \cdot N_H \cdot K_W \cdot K_H \cdot \tfrac{1}{S_W \cdot S_H}, \tag{13}$$

where $C_{\text{in}}$ and $C_{\text{out}}$ are the number of input and output channels, $(N_W, N_H)$ is the size of the input, $(K_W, K_H)$ is the size of the kernel, and $(S_W, S_H)$ is the stride value. The BOPs count is then

$$\text{BOPs(conv)} = \text{MAC(conv)} \cdot b_w \cdot b_a, \tag{14}$$

where $b_w$ and $b_a$ denote the number of bits used for weight and activations.

As described in section 3, the Haar transform is separable between the input channels, and can be viewed as four $2 \times 2$ convolutions with stride $(2, 2)$ and binary weights. Hence, the one-level transform requires $4 \cdot C_{\text{in}} \cdot W \cdot H \cdot b_a$ BOPs. The transform can be used with more levels of compression explained in section 3, on down-scaled inputs, resulting in a total of

$$\sum_{l=1}^{L} 4 \cdot C_{\text{in}} \cdot N_W \cdot N_H \cdot \tfrac{1}{4^{l-1}} \cdot b_a \tag{15}$$

BOPs, where $L$ is the level of compression. Similarly, the inverse-transform result in the same calculation, only with $C_{\text{out}}$ in place of $C_{\text{in}}$. To demonstrate the relatively small cost of the compression, consider a $1 \times 1$ convolution with $C_{\text{in}} = 160$, $C_{\text{out}} = 960$, input size of $(34, 34)$, and quantization $b_w = b_a = 8$ (which is part of a network used in section 5). This layer costs $11,364M$ BOPs. Using a 3 levels wavelet transform and its inverse for this layer results in $54M$ BOPs, a negligible cost which allows for better compression, as we demonstrate next.

# E  Full Segmentation Results

Table 6 shows the performance and BOPs of each model trained by us in the experiments described in subsection 5.3. In our experience, quantizing with 4bit activations resulted in a sharp drop in results. While other more sophisticated methods experience less of a decline, it is still significant. Using said methods for 8bit with our approach will also result in improved scores for WCC.

Table 6: Validation results for semantic segmentation task using DeepLabV3plus with MobileNetV2 as the backbone. Segmentation performance is measured by mean intersection over union (mIoU)

| Precision (W/A) | Wavelet shrinkage | Cityscapes BOPs (B) | mIoU | Pascal VOC BOPs (B) | mIoU |
|---|---|---|---|---|---|
| FP32 | None | 36,377 | 0.717 | 4,534 | 0.715 |
| 8bit / 8bit | None | 2,273 | 0.701 | 283 | 0.712 |
| 8bit / 6bit | None | 1,705 | 0.683 | 212 | 0.678 |
| 8bit / 4bit | None | 1,136 | 0.173 | 141 | 0.095 |
| 8bit / 8bit | 50% | 1,213 | 0.681 | 150 | 0.675 |
| 8bit / 8bit | 25% | 673 | 0.620 | 82 | 0.611 |
| 8bit / 8bit | 12.5% | 403 | 0.552 | 48 | 0.519 |
| 4bit / 8bit | None | 1,136 | 0.682 | 141 | 0.675 |
| 4bit / 6bit | None | 852 | 0.669 | 106 | 0.657 |
| 4bit / 4bit | None | 568 | 0.190 | 70 | 0.099 |
| 4bit / 8bit | 50% | 616 | 0.667 | 76 | 0.661 |
| 4bit / 8bit | 25% | 346 | 0.621 | 42 | 0.583 |
| 4bit / 8bit | 12.5% | 211 | 0.549 | 24 | 0.515 |

# F   WCC with Different Wavelets

When considering different wavelets for compression, the added computational cost should also be weighted. Calculating the MAC operations for the transform as explained in Appendix D, a $2k \times 2k$ kernel represented in $b_w$ bits results in $k^2 b_w$ times the BOPs for the same input compared to the Haar transform. Table 7 compares different WCC layer configurations using several wavelets for Cityscapes semantic segmentation task, $b_w$ was set to 32-bit floating-point for all options.

Table 7: Validation results for semantic segmentation task using DeepLabV3plus with MobileNetV2 as the backbone. All experiments are using 8bit/8bit quantization.

| Wavelet Type | Filter Size | 50% Shrinkage mIoU | 25% Shrinkage mIoU | 12.5% Shrinkage mIoU |
|---|---|---|---|---|
| Haar | $2 \times 2$ | 0.681 | 0.620 | 0.552 |
| Daubechies 2 (db2) | $4 \times 4$ | 0.680 | 0.630 | 0.561 |
| Daubechies 3 (db3) | $6 \times 6$ | 0.678 | 0.629 | 0.560 |
| Coiflets 1 (coif1) | $6 \times 6$ | 0.676 | 0.637 | 0.562 |
| Biorthogonal 1.3 (bior1.3) | $6 \times 6$ | 0.684 | 0.637 | 0.564 |
| Biorthogonal 2.2 (bior2.2) | $6 \times 6$ | 0.677 | 0.638 | 0.566 |
| Symlets 4 (sym4) | $8 \times 8$ | 0.675 | 0.629 | 0.565 |

# G   Depth Prediction Qualitative Results

Qualitive results for subsection 5.4 are presented in Figure 6.

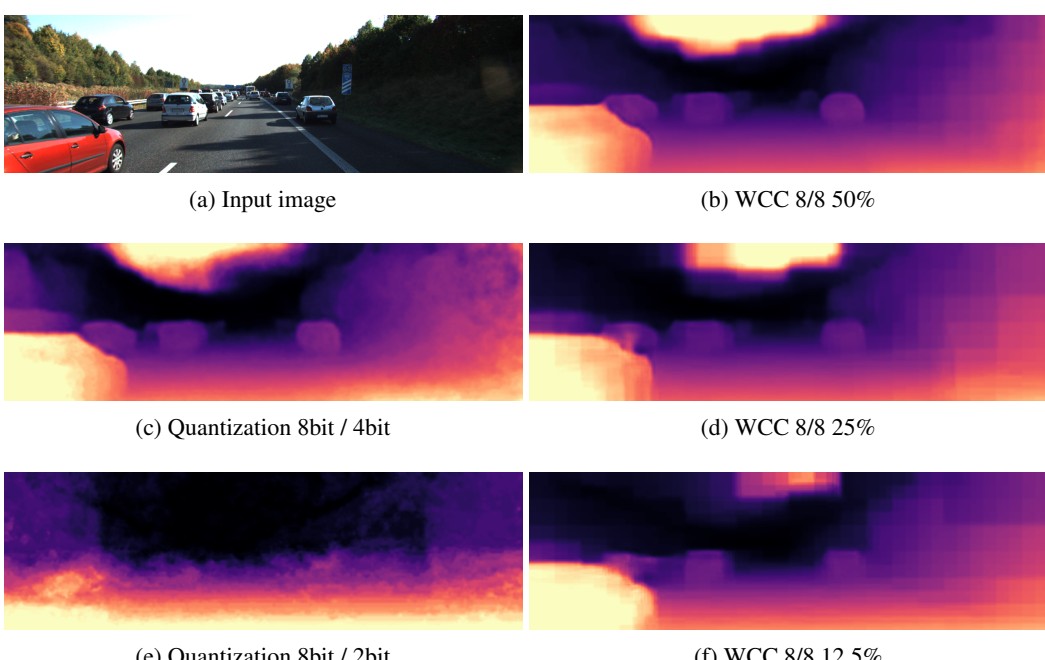

(a) Input image

(b) WCC 8/8 50%

(c) Quantization 8bit / 4bit

(d) WCC 8/8 25%

(e) Quantization 8bit / 2bit

(f) WCC 8/8 12.5%

Figure 6: Kitti depth estimation prediction examples on Monodepth2. All networks use weight quantization of 8-bits. (c), (e) show results for activation quantization of 4bit and 2bit respectively. (b), (d), (f) show results for WCC with 50%, 25% & 12.5% compression factor respectively.

# H Inference Times on a GPU using a Custom Implementation

In this paper we followed the common practice of measuring the theoretical speedup of our approach in terms of BOPs, as commonly done in quantization works (e.g., [67]). That is because our aim is to speed up inference times on low-resource devices where BOPs are the main computational bottleneck. However, WCC includes several operations done in tandem which are not straightforward to implement efficiently using existing CNN frameworks. These include the forward and inverse Haar transforms, gather and scatter operations using a single index list, and top-k selection on the pixel-wise norms across channels. To demonstrate that our approach can be effective in practice on typical GPUs, we also developed a custom CUDA implementation for the ingredients of WCC. On top of that, having a custom implementation allows for several opportunities to further speed up the process, and keep the memory bandwidth low in certain common scenarios, as we detail below.

The key ingredients of our custom implementation are as follows:

1. An in-place implementation of the forward and inverse Haar transforms, using a single memory read and write for all levels. This results in an inference time for the transforms, which is comparable to double the one of average pooling.

2. A custom gather and scatter kernels that use a single index list for all channels.

3. A kernel for a fused depthwise convolution and Haar transform. Since our framework rely on the idea of separable depthwise convolutions, it is natural to fuse together consecutive separable operations like depthwise convolution and the Haar transform that typically follows it. This saves memory read and write, as well as several computations that are joint for both operations (because both are separable).

To demonstrate the effectiveness of our implementation, we compare the inference time of an inverted bottleneck residual block used in modern architectures like MobileNets [50, 27], ConvNext [42], and EfficientNets [52, 53]. The inverted residual block that we test for timing purposes reads

$$\mathbf{x}^{(l+1)} = \mathbf{x}^{(l)} + \mathbf{K}_{1\times1}^{l_3} \left( \mathbf{K}_{dw}^{l_2} \left( \mathbf{K}_{1\times1}^{l_1} \mathbf{x}^{(l)} \right) \right), \tag{16}$$

where $\mathbf{K}_{1\times1}^{l_3}, \mathbf{K}_{1\times1}^{l_1}$ are $1 \times 1$ convolutions and $\mathbf{K}_{dw}^{l_2}$ which is typically applied on a much larger channel dimension than that of $\mathbf{x}^{(l)}$ (the factor between the channel sizes if often called *expansion*). We note that for the purpose of timings, we omit the non-linear activations which are typically fused into the convolution kernels are are applied at negligible cost, if simple. The timings were obtained using PyTorch, that is bounded to CUDA kernels using the package `ctypes`, and is run on an NVIDIA 1080ti GPU on an isolated Linux machine. All runs are applied using the maximal batch size possible, and are averaged over 100 trials.

Table 8 summarizes the results for different parameters. It is clear, as expected, that the speedup is better for lower shrinkage rates, and when the number of channels is higher. The latter is a key theoretical aspect of the speedup - the complexity of $1 \times 1$ convolutions is quadratic in the number of channels, while the complexity of the WCC additional operations is linear. Hence, we expect more speedup as the number of channels grows in the future. We would like to stress that (1) our implementation can probably be further optimized and (2) server GPUs may be far from the typical prototype low-resource edge device in common scenarios.

**Memory bandwidth and traffic**: The most complicated and optimized operation of CNNs is the dense matrix-matrix multiplication, i.e., the $1 \times 1$ convolution. Typically, a tile (part of an image) from *all channels* has to be read by each group of threads to compute a tile of an output channel. That is in addition to reading the relevant weights. In our work, we ease these memory reads by simply reducing the dimensions of the feature maps. The other operations in the network are separable (activations, Haar, gather/scatter, depthwise convolutions) and are of linear complexity in their memory reads. Hence, as more channels are used, the relative memory traffic using WCC compared to a standard convolution will decrease. Furthermore, considering a common inverted residual block as in (16) with a large expansion, the peak memory lies in the input and output of $\mathbf{K}_{dw}^{l_2}$. Using our method, we may apply the Haar transforms, depthwise convolutions, and scatter/gather separately and in parts using relatively small intermediate allocated memory and write the full result in a compressed form. This way, we only store the full result in a compressed manner towards the input of $\mathbf{K}_{1\times1}^{l_3}$, which needs to be complete before the $1 \times 1$ convolution. Other scenarios for memory savings can be obtained for different architectures, devices, and scenarios.

Table 8: Inference timing results.

| Image size | $c_{in}$ | Expansion | Batch | Comp. rate | Standard [s] | Ours [s] | Speedup |
|---|---|---|---|---|---|---|---|
| $96 \times 96$ | 512 | 2 | 48 | 0.25 | $1.82 \cdot 10^{-1}$ | $1.19 \cdot 10^{-1}$ | $\times 1.52$ |
| $96 \times 96$ | 512 | 2 | 48 | 0.5 | $1.82 \cdot 10^{-1}$ | $1.52 \cdot 10^{-1}$ | $\times 1.20$ |
| $96 \times 96$ | 512 | 4 | 24 | 0.25 | $1.78 \cdot 10^{-1}$ | $1.04 \cdot 10^{-1}$ | $\times 1.71$ |
| $96 \times 96$ | 512 | 4 | 24 | 0.5 | $1.76 \cdot 10^{-1}$ | $1.36 \cdot 10^{-1}$ | $\times 1.29$ |
| $96 \times 96$ | 1024 | 2 | 12 | 0.25 | $1.38 \cdot 10^{-1}$ | $7.11 \cdot 10^{-2}$ | $\times 1.94$ |
| $96 \times 96$ | 1024 | 2 | 12 | 0.5 | $1.38 \cdot 10^{-1}$ | $1.00 \cdot 10^{-1}$ | $\times 1.38$ |
| $96 \times 96$ | 1024 | 4 | 6 | 0.25 | $1.36 \cdot 10^{-1}$ | $6.36 \cdot 10^{-2}$ | $\times 2.13$ |
| $96 \times 96$ | 1024 | 4 | 6 | 0.5 | $1.36 \cdot 10^{-1}$ | $9.20 \cdot 10^{-2}$ | $\times 1.47$ |
| $128 \times 128$ | 128 | 4 | 80 | 0.25 | $1.67 \cdot 10^{-1}$ | $1.39 \cdot 10^{-1}$ | $\times 1.20$ |
| $128 \times 128$ | 128 | 6 | 64 | 0.25 | $1.99 \cdot 10^{-1}$ | $1.56 \cdot 10^{-1}$ | $\times 1.28$ |
| $128 \times 128$ | 256 | 4 | 32 | 0.125 | $1.65 \cdot 10^{-1}$ | $0.91 \cdot 10^{-2}$ | $\times 1.81$ |
| $128 \times 128$ | 256 | 4 | 32 | 0.25 | $1.66 \cdot 10^{-1}$ | $1.08 \cdot 10^{-1}$ | $\times 1.53$ |
| $128 \times 128$ | 256 | 4 | 32 | 0.5 | $1.66 \cdot 10^{-1}$ | $1.31 \cdot 10^{-1}$ | $\times 1.27$ |
| $256 \times 256$ | 256 | 4 | 8 | 0.125 | $1.72 \cdot 10^{-1}$ | $9.27 \cdot 10^{-2}$ | $\times 1.85$ |
| $256 \times 256$ | 256 | 4 | 8 | 0.25 | $1.72 \cdot 10^{-1}$ | $1.09 \cdot 10^{-1}$ | $\times 1.57$ |
| $256 \times 256$ | 256 | 4 | 8 | 0.5 | $1.69 \cdot 10^{-1}$ | $1.31 \cdot 10^{-1}$ | $\times 1.29$ |
| $256 \times 256$ | 256 | 8 | 4 | 0.125 | $1.68 \cdot 10^{-1}$ | $8.45 \cdot 10^{-2}$ | $\times 1.99$ |
| $256 \times 256$ | 256 | 8 | 4 | 0.25 | $1.68 \cdot 10^{-1}$ | $1.00 \cdot 10^{-1}$ | $\times 1.68$ |
| $256 \times 256$ | 256 | 8 | 4 | 0.5 | $1.68 \cdot 10^{-1}$ | $1.21 \cdot 10^{-1}$ | $\times 1.39$ |
| $512 \times 512$ | 128 | 6 | 4 | 0.25 | $2.13 \cdot 10^{-1}$ | $1.51 \cdot 10^{-1}$ | $\times 1.41$ |
| $512 \times 512$ | 128 | 6 | 2 | 0.5 | $1.04 \cdot 10^{-1}$ | $9.21 \cdot 10^{-2}$ | $\times 1.12$ |
| $512 \times 512$ | 256 | 4 | 2 | 0.125 | $1.74 \cdot 10^{-1}$ | $9.85 \cdot 10^{-2}$ | $\times 1.77$ |
| $512 \times 512$ | 256 | 4 | 2 | 0.25 | $1.73 \cdot 10^{-1}$ | $1.14 \cdot 10^{-1}$ | $\times 1.52$ |
| $512 \times 512$ | 256 | 4 | 2 | 0.5 | $1.72 \cdot 10^{-1}$ | $1.43 \cdot 10^{-1}$ | $\times 1.20$ |