# OpenReview forum: "Wavelet Feature Maps Compression for Image-to-Image CNNs"
_NeurIPS.cc/2022/Conference — NeurIPS 2022 Accept_

### Official Review · Reviewer_Jstv · 2022-07-11

**Rating:** 5
**Confidence:** 3
**Soundness:** 3 good
**Presentation:** 3 good
**Contribution:** 3 good

**Summary:**

The paper studies the network quantization and proposes a wavelet compressed convolution for the high-resolution activation maps. Experiments show that the proposed approach achieves compression rates equivalent to 1-4bit activation quantization with relatively small.

**Questions:**

see weaknesses

**Ethics Review Area:**

["I don’t know"]

**Limitations:**

I do not see too many limitations.

**Strengths And Weaknesses:**

**Strengths**

1. This paper introduces a wavelet compressed convolution to reduce the required computational cost and memory usage for CNNs under high-resolution-based feature maps.
2. The paper tries to solve a practical problem. The memory usage and the computational cost are quite high when the activation map is under high resolution.
3. The idea is interesting. The authors introduce a Haar-wavelet transform into the convolution process to reduce the computational cost, instead of directly quantizing the feature map. This direction is interesting to explore further.

**Weaknesses**

1. The experimental results are not sufficient to support the author's claim. The authors claim their method can reduce memory usage. However, the reviewer does not see such a report on memory usage after compression.
2. It is better to further polish the introduction section. In the current version, it is hard to identify the technical contribution of this paper.
3. It is better to choose another baseline method or make the current baseline method stronger for comparison as the reproduction of the baseline method in this paper is much worse than the reported results in [43]. For example, in Table 1, [43] has 35.34 mAP around 280 BOPs. However, the author's reproduction is only 31.44. If the baseline method is worse, it is easier to achieve a smaller performance drop after compression.
4. Can the Haar-wavelet transform be used to the CNN without quantization. It seems that this method can also be used for full-precision CNNs (correct me if I am wrong). The reviewer does not see the reason why it is necessary to further quantize the model after the Haar-wavelet transform. Why not directly compare the BOPs after the Haar-wavelet transform.

---

> ### Author Response · Authors · 2022-08-02
> **Response to Reviewer Jstv**
>
> We thank the reviewer for the constructive review.
>
> 1) **Regarding memory**: Indeed, we did not fully demonstrate this point in the paper. In lines 221-224 we explain that the one operation that is difficult in terms of memory reads is the $1\times1$ convolution, where typically, a tile (part of an image) from all channels has to be read to compute a tile of the output channel. That is in addition to reading the relevant weights. In our work, we ease these operations by reducing the dimensions of the feature maps. The other operations in the network are separable (activations, Haar, gather/scatter, depthwise convolutions) and are linear in their memory reads. I.e., a single input is read for each output. Hence, our method is most beneficial for large networks with many channels. Beyond this, please see **Appendix I** for a discussion about memory in inverse-bottleneck networks. In our experience, the speedups we obtained are because of fewer memory reads in the inverse bottleneck part.
>
> 2) **Regarding introduction**: We will do our best to clarify the technical contribution in the revised version, and if place permits, we will add a "key contributions" paragraph.
>
> 3) **Regarding baseline**: First, we would like to mention that Nagel et al. (2021) have no published code, and we could not reproduce their results. Second, the reviewer is correct in pointing out that it is easier to achieve a smaller performance drop when the baseline method is worse. However, in the same table, when compressing the model to 198.5B BOPs (and even 155.4B BOPs) using our method, it achieves superior results to the reported results of Nagel et al.'s (2021) 4bit/4bit (185.8B BOPs). Therefore, achieving the best **absolute** score for this compression rate while using an inferior baseline and training procedure.
>
> 4) **Regarding WCC without quantization**: It is correct that WCC can be used without quantization at all. While performing well in terms of results, this will not be competitive with quantization in terms of compression rates (e.g., BOPs). However, we show that our compression can be easily incorporated together with a moderate quantization to achieve superior results to aggressive quantization. Please refer to the additional experiment we provide, in which we include a full-precision compression demonstration.

---

### Official Review · Reviewer_m652 · 2022-07-11

**Rating:** 5
**Confidence:** 4
**Soundness:** 3 good
**Presentation:** 3 good
**Contribution:** 3 good

**Summary:**

This paper proposes to incorporate wavelet decomposition and compression for quantization in image-to-image tasks.

**Questions:**

See the question in Weakness

**Limitations:**

Yes

**Strengths And Weaknesses:**

Strengths
1. It proposes Wavelet Compressed Convolution (WCC) for high-resolution activation map compression, which is interesting.
2. The proposed method dramatically improves the results over aggressive quantization for the same compression rates while retaining the baseline network architecture.

Weaknesses
1. The appendix should be put in the supplementary files instead of the main paper
2. What are the top-k/discarded components of wavelet compositions? Are the discarded ones the high-frequency components?
3. If the discarded ones are high-frequency components, the use of wavelet decomposition and choice of top-k entries will lead to detail loss. It may be effective for image-to-image tasks, such as depth estimation and segmentation, where the detailed textures are not important (but the sharp edges are important). What if for the super-resolution task with high-resolution inputs?
4. The proposed method can only be applied in 1x1 convolutions, largely limiting its application.
5. For each feature, the proposed method performs the wavelet composition, top-k choose, and gather operation, which will interrupt the calculation flow of GPU. I doubt such a process will slow down the actual inference time, as the original GPU calculation flow is much smoother. The authors are supposed to provide the actual running time.

---

> ### Author Response · Authors · 2022-08-02
> **Response to Reviewer m652**
>
> We thank the reviewer for the constructive review.
>
> 1) **Regarding separated appendix**: According to the FAQ for Authors (https://nips.cc/Conferences/2022/PaperInformation/NeurIPS-FAQ), appendices are allowed to be included with the main submission file. We believe it makes the appendix easier to access, but have no objection to separate it from the main paper if required.
>
> 2) **Regarding discarded components**: While the discarded components are typically in the output of the high-pass filters, those outputs are known to be sparse since images are typically piece-wise smooth. Edges generate responses in the high-pass filters, and if are high, those responses are kept and details are preserved. That is, in fact, part of how JPEG2000 compresses images at a minimal quality degradation.
>
> 3) **Regarding super-resolution**: When compressing the network, a degradation in performance is to be expected. Even if the discarded components are from the outputs of the high-pass filters, they are close to $0$ in magnitude and contribute little to the convolution result. To demonstrate it, we provide an additional experiment, compressing the popular super-resolution network EDSR (Lim et al. 2017) with the DIV2K dataset (Agustsson et al. 2017).
>
> 4) **Regarding $1\times 1$ convolutions**: As explained in **Appendix A**, in most cases it is simple and common to replace a $k\times k$ convolution with a set of two convolutions, one depthwise-$k\times k$, and one pointwise-$1\times 1$. This is an effective approach that alone provides compression with little-to-no impact on the results. Please refer to the additional experiment that we provide, where we demonstrate it for another application (super-resolution) suggested in the previous point.
>
> 5) **Regarding inference times**: Please refer to the general comment and new **Appendix I** regarding inference times and implementation details. It is true that the PyTorch implementation of our method indeed has a complex flow on GPUs, and is not faster than the standard convolutions. However, using our custom implementation, we were able to speed up the computation on a GPU. We would like to stress that (1) our implementation can probably be further optimized by merging kernel calls and other types of optimizations (we saw that each kernel call has a significant overhead and generally we do not consider ourselves as expert CUDA programmers), and (2) server GPUs are far from the prototype low-resource edge-device that the community has in mind. Our timing results just serve as a proof of concept.

---

> > ### Author Response · Authors · 2022-08-08
> > **Response to reviewer m652**
> >
> > Dear Reviewer m652,
> >
> > We thank you again for your review.
> >
> > We hope our response clarified the points raised in your review. Since the discussion period is close to the end, we would be glad to know if you feel that our rebuttal response has addressed your questions/concerns. Should more clarifications or additional information be needed, we are ready and willing to do so. Please kindly let us know your feedback as we feel that we have adequately addressed your concerns (e.g., super-resolution experiment, inference timings, and also point 4), and if indeed so we sincerely ask you to consider raising our scores to reflect that.
> >
> > We appreciate your time and thought.
> >
> > Sincerely,
> >
> > The authors.

---

### Official Review · Reviewer_QigK · 2022-07-11

**Rating:** 6
**Confidence:** 4
**Soundness:** 3 good
**Presentation:** 3 good
**Contribution:** 3 good

**Summary:**

This paper newly suggests feature map compression methods for Image-to-Image CNNs, while most of the compression methods focus on classification models. The authors utilize wavelet transform to adaptively extract spatially meaningful pixels and perform pointwise convolution(=conv1x1) on the sparse feature map. Combining with light 8-bit quantization, it shows outstanding results compared to previous methods on various image-to-image vision tasks.

**Questions:**

- Although pointwise convolution is performed in the sparse feature map, because selected pixels are not contiguous, the computational boost might not be noticeable. Also, it has to zero-fill the output after pointwise conv1x1 to perform inverse wavelet transform which requires memory for the full feature map anyway in the end. How was the inference time compared to the quantization-only method?

- ~~If only BOP matters (and memory doesn't matter for the evaluation), have you tried residual filling (or the same filling = just reuse non-top k entries for inverse transform) rather than zero-filling?~~

- Depending on the bias of conv1x1 kernel, the output from zero input can be far from zero. In this case, zero-filing can be harmful. Though in most cases, pixelwise operation use bias=False option, some might use bias for conv1x1 operations. How can your algorithm deal with such case?

- Notation for the concatenation in Eq.(6) is confusing, since the dimension of $y_1^2$ and $y_2^1$ are different ($C\times(H/4)\times(W/4)$ and $C\times(H/2)\times(W/2)$). Since 'concatenate' means channel-wise concatenation in many literatures.
specifying the orientation of concatenation and the dimension of concatenated vector $y$ would be helpful.

- How does the distribution of feature norm look like before the joint shrinkage? Depending on the number of features with near zero norm, this quantization technique could be either very harmful or effective.

**Limitations:**

- Dense prediction models that do not use conv1x1 operation can not adopt this quantization technique.


**Strengths And Weaknesses:**

Strength
- Problem setting: This paper clearly states the limitation of previous methods (mainly focused on the classification model and not effective enough for image-to-image CNNs) and suggests a proper algorithm to solve the given problem.
- Robustness: Since this method is simple yet powerful, it can be easily adopted to most types of Image-to-Image CNN architectures.
- Novelty: Utilizing the equivalence of conv1x1 operation, selecting top-k pixels on spatially-concatenated feature map in wavelet domain is impressive.

Weakness
- Lack of comparison with previous methods: Since it is more of a pixel-wise compression method, rather than a quantization method. It would be better to add a comparison with other efficient (or light) models in Image-to-Image domains. For example, PointRend is a segmentation model that also adaptively selects the point features using a tiny prediction model for efficient spatially-adaptive operation. Also, a comparison with other quantization methods seems missing.

---

> ### Author Response · Authors · 2022-08-02
> **Response to Reviewer QigK**
>
> We thank the reviewer for the constructive review.
>
> - **Regarding other quantization methods and PointRend**: Our quantization method is uniform, and hence most efficient to implement in hardware. Other more sophisticated quantization methods can be utilized within our framework, for quantizing the wavelet features after shrinkage and may provide another upgrade to the accuracy. Moreover, PointRend, as far as we understand, suggests a network head to be used for prediction of segmentation maps given features that are generated by a backbone. Our method is actually orthogonal to PointRend, as it is mostly concerned with compressing the backbone which can be significant computationally. The two methods can be used together to achieve the best results. We will cite PointRend in our revised version.
>
> - **Regarding inference times**: Please see the general comment as well as the results and discussion in Appendix I regarding inference times and memory usage. We would like to stress that (1) our implementation can probably be further optimized by merging kernel calls and other types of optimizations (we saw that each kernel call has a significant overhead and generally we do not consider ourselves as expert CUDA programmers), and (2) server GPUs are far from the prototype low-resource edge-device that people in the community have in mind. Our timing results just serve as a proof-of-concept.
>
> - **Regarding other types of filling**: To maintain the equivalence discussed in Eq. (11)-(12), we used the zero-filling, which, as far as we know is the standard way to apply wavelet compression. We do not entirely follow the reviewer's meaning here, but other filling approaches can definitely be considered if they lead to the same computational savings. We will be happy to further discuss this direction, and consider it for future work.
>
> - **Regarding bias=True**: As the reviewer mentions, most works use bias=False. When considering bias=True, one approach is to add the bias after the inverse transform. Please refer to the additional experiment we provide, in which we use this solution with no impact on results. Another approach is to add the bias only to the low-passed transform features (a small block that is typically not zeroed), but this can be done only using a custom implementation.
>
> - **Regarding concatenation in Eq. 6**: The concatenation is done in the spatial domain after flattening each of the vectors. This is obtained for each channel separately. We will clarify this point in the revised version. We note that in our new custom implementation, the transform is done in-place in tiles without concatenation. The description in the paper describes the PyTorch implementation using CNN tools.
>
> - **Regarding the distribution of feature norms**: The main assumption of wavelet compression methods is that images are piece-wise smooth and therefore can be sparsely represented in the wavelet domain. Here, we essentially  demonstrate that this is approximately true for intermediate feature maps of image-to-image networks, and also that there is a high correlation in the edge locations of different channels, as was demonstrated in other works (e.g., GhostNet.). Therefore, the shrinkage (index selection) can be applied jointly for all channels while maintaining low error (e.g., Fig. 1). Still, since those assumptions hold only approximately, our compression rate, which is 12.5\%-50\%, is relatively higher than the typical compression rate that is used for a single natural image.
>
> - **Regarding limitation to 1X1 convs**: As explained in **Appendix A**, in most cases it is simple and common to replace a $k\times k$ convolution with a set of two convolutions, one depthwise-$k\times k$, and one pointwise-$1\times 1$. This is a very effective approach that alone provides compression with little-to-no impact on the results. Please refer to the additional experiment that we provide, where we demonstrate it for another application (super-resolution) suggested by the reviewers.

---

> > ### Comment · Reviewer_QigK · 2022-08-08
> > **Feedback**
> >
> > The authors have addressed most of my concerns.
> >
> > Though the improvement of inference time is rather theoretical and the compression rate is relatively high compared to typical approaches, this approach is not only novel but also could be applied to various Image-to-Image CNN tasks orthogonally. I believe this work can positively contribute to the vision community.
> >
> > Therefore, I change my rating from 5 to 6.

---

### Official Review · Reviewer_385X · 2022-07-11

**Rating:** 6
**Confidence:** 4
**Soundness:** 4 excellent
**Presentation:** 4 excellent
**Contribution:** 3 good

**Summary:**

Wavelet Compressed Convolution (WCC) is introduced which addresses the problem of compressing CNNs to reduce memory and runtime requirements. WCC is motivated by the fact that "aggressive" quantization (fp32 -> int4 or smaller) often works well for classification problems but not for dense prediction tasks like image-to-image translation models. WCC solves this problem by applying a wavelet transform to the activation maps within a model. This has the effect of increasing sparsity which is utilized by mapping the large, sparse tensor to a compact representation holding the top-k largest values along with their location. A 1x1 conv is then applied to this smaller representation (the authors show that the wavelet transform and 1x1 conv commute), which is much cheaper. The wavelet transform is then inverted to recover the (approximate) activations.

WCC is empirically evaluated on several tasks (object detection, semantic segmentation, and monocular depth estimation) where the experiments show that WCC outperforms naive quantization at equivalent bit depths (e.g. 2-bit quantization = 8-bit quantization + WCC that compresses to 25% of the original size).

**Questions:**

As mentioned in the strengths and weakness section:

Why are the results for "our baseline + WCC" with "wavelet shrinkage = None" worse than the published results? I think this is ok since any improvement in the cases with WCC and lower quantization levels is even more impressive, but a close baseline strengthens the results as well as confidence in the implemented model.


**Limitations:**

Yes. The authors explain that WCC does not work well for small feature maps at the end of Section 4.

**Strengths And Weaknesses:**

Strengths:
1. The authors make an important observation that quantization methods that work for classification often don't work for dense prediction tasks. The proposed method (wavelet transformation and top-k selection followed by 1x1 conv) is conceptually simple, well-motivated, and appears to work well in practice when more extreme quantization levels are needed (less than 4-bits per value).

Weaknesses:
1. WCC often doesn't work as well as naive quantization at moderate quantization levels, though this may be due to baseline models that don't reach the performance of published results (e.g. "4bit / 8 bit None" in Tables 1 and 2).

2. The work immediately raises the question of why the transform (Haar was used in the paper) isn't learned.

---

> ### Author Response · Authors · 2022-08-02
> **Response to Reviewer 385X**
>
> We thank the reviewer for the constructive review.
>
> 1) **Regarding "our baseline + WCC" worse than published results**:  Nagel et al. (2021) have no published code and we couldn't reproduce their results. In Tables 1 and 2, our baseline (4bit/8bit/None) is based on a uniform quantization version in the work of Li et al. (2020). It is important to note that "shrinkage = None" means **not using WCC at all**, but using quantization only. While explained in lines 271-273, we see how this can be confusing and will further clarify that in both the text and the table in the revised version.
>
> 2) **Regarding why Haar isn't learned**: As described in Section 4 (lines 174-177), and as can be seen in Eq. 3, the Haar wavelet can be implemented using additions and subtractions only, and therefore more efficient. Any type of wavelet can be used within our framework - even a learned one, but one should consider the added computational cost. **Appendix F** demonstrates the use of other wavelets (albeit not learned ones) and discusses the additional computational cost.

---

> > ### Comment · Reviewer_385X · 2022-08-08
> > **authors' responses satisfy my concerns**
> >
> > 1. It's unfortunate that the results from Nagel 2021 couldn't be reproduced, but it's also a common and understandable situation.
> >
> > 2. Using the Haar wavelet due to a focus on computational complexity makes sense, and Appendix F does give a sense of the mIoU gains achievable with more complex wavelets. Adding a row for a learned transform would further strengthen the paper but isn't required for acceptance in my opinion.
> >
> > Thank you to the authors for responding to my questions / concerns. I think my original rating of 6 (weak accept) holds.

---

### Author Response · Authors · 2022-08-02
**General comment for the AC and all the reviewers**

We thank all the reviewers for their thoughtful and constructive review of our paper.
To relieve some concerns raised by the reviewers, we provide an additional experiment with the EDSR architecture (Lim et al., 2017) for the super-resolution task. The details and the results of the experiment are presented in full in **Appendix H** in the rebuttal revision of the paper.

- We hope that the results of this experiment relieve Reviewer **m652**'s concern regarding the loss of details for the super-resolution task.
- Since EDSR originally includes $3\times 3$ convolutions only, we utilize the technique described in Appendix A and convert the model to have pointwise and separable convolutions, which are cheaper to apply in terms of BOPs, with no empirical degradation. We hope this relieves Reviewers **QigK** and **m652**'s concern regarding this subject.
- All the convolutions in the original model use bias=True. Instead of adding the bias during the convolution, we add it after the inverse wavelet transform. This relieves Reviewer **QigK**'s concern regarding using WCC when the convolution has bias.
- Reviewer **Jstv** suggested that our method does not have to rely on quantization and can be applied with full precision. That is true, and we added full-precision results to demonstrate it. However, we found that having a good balance between quantization and the WCC compression rate reduces the computation substantially with little impact on the results.

In addition, we would also like to address the comments on inference times brought up by reviewers **QigK** and **m652**. Generally, in this paper, we follow the common practice of quantization works, which usually apply full-precision computations in practice and only simulate low-bit quantized computations. Like such works, we measure the theoretical performance of our compression method in terms of bit-operations (BOPs), which are widely used in the literature. Furthermore, the aim of such works, and ours, is typically to improve inference time and power consumption on low-resource edge devices, where BOPs (or FLOPs) may indeed be the main bottleneck. The actually gained speedup of a particular approach may highly depend on the hardware at hand. Still, to address the concerns, we developed a custom CUDA implementation for WCC to demonstrate that the entire process can be obtained efficiently on our GPU. The results, including discussions, now appear in **Appendix I**, and more specific answers are given to the reviewers separately.

We will further address individual concerns in a personal response to each reviewer.

With kind regards,
The authors.

---

> ### Author Response · Authors · 2022-08-02
> **EDSR compression results**
>
> For easier access, we also provide the EDSR compression results here, for the task of 2x super-resolution.
>
> | Precision (W / A) | Wavelet Shrinkage | BOPs (B)       | PSNR   |
> |------------------------|---------------------|--------------------|------------|
> | Full precision*     | None                 | 975,838         | 35.03   |
> | Full precision      | None                 | 123,674         | 35.02   |
> | | | |
> | Full precision      | 50%                  | 70,017           | 34.98   |
> | Full precision      | 25%                  | 42,910           | 34.93   |
> | Full precision      | 12.5%               | 29,357           | 34.76   |
> | | | |
> | 8bit / 16bit           | None                | 15,459           | 34.55  |
> | | | |
> | 8bit / 8bit             | None                | 7,730              | 34.53  |
> | 8bit / 4bit             | None                | 3,865              | 34.49  |
> | | | |
> | 8bit / 16bit           | 50%                  | 8,961             | 34.55  |
> | | | |
> | 8bit / 8bit             | 50%                 | 4,480              | 34.53  |
> | 8bit / 8bit             | 25%                 | 2,768              | 34.50  |
>
> _The asterisk * denotes the original network, without applying separable convolutions_
>
> The authors.

---

### Comment · Area_Chair_dYVh · 2022-08-07
**Discussion period**

Thank you to all the reviewers for the great effort in reviewing the paper and the authors for the responses.

As the author-reviewer discussion period is almost over, I want to ensure that reviewers have read the authors' responses and engage with the authors if needed.

If you haven't done this, could you please take a moment to read through the authors' responses, update the reviews to indicate that you have read the authors' responses, or communicate with the authors if needed? You can also share in private conversations with the reviewing team.

Please continue to share your thoughts. Thank you!

---

### Meta-Review · Area_Chair_dYVh · 2022-08-23

**Recommendation:** Accept
**Confidence:** Less certain

**Metareview:**

The paper proposes a method for compressing feature maps in convolutional neural networks to reduce the computational cost. The method is tailored to image-to-image networks, where existing compression schemes do not work well.

After the rebuttal, all reviewers support the publication of the manuscript. The reviewers note that the problem setting is important (i.e., compression for image-to-image networks, where existing compression schemes do not work well as identified by the paper under review), and that the proposed method works well and is interesting. The reviewers also identified a few weaknesses that for the most part have been cleared up by the author's response. I, therefore, recommend accepting the paper.


**Award:**

No

---

### Decision · Program_Chairs · 2022-09-14

Accept